# A Feature-Level Point Cloud Fusion Method for Timber Volume of Forest Stands Estimation

**Lijie Guo** [1,2,3], **Yanjie Wu** [1,2,3], **Lei Deng** [1,2,3,*], **Peng Hou** [4], **Jun Zhai** [4] **and Yan Chen** [4]

1   College of Resource Environment and Tourism, Capital Normal University, Beijing 100048, China; 2210901025@cnu.edu.cn (L.G.); 2210902132@cnu.edu.cn (Y.W.)
2   Engineering Research Center of Spatial Information Technology, Ministry of Education, Capital Normal University, Beijing 100048, China
3   Beijing Laboratory of Water Resources Security, Capital Normal University, Beijing 100048, China
4   Satellite Application Center for Ecology and Environment, Ministry of Ecology and Environment, Beijing 100094, China; houpcy@163.com (P.H.); zhaij@lreis.ac.cn (J.Z.); chenyan30033@163.com (Y.C.)
*   Correspondence: denglei@cnu.edu.cn

**Abstract:** Accurate diameter at breast height (DBH) and tree height (H) information can be acquired through terrestrial laser scanning (TLS) and airborne LiDAR scanner (ALS) point cloud, respectively. To utilize these two features simultaneously but avoid the difficulties of point cloud fusion, such as technical complexity and time-consuming and laborious efforts, a feature-level point cloud fusion method (FFATTe) is proposed in this paper. Firstly, the TLS and ALS point cloud data in a plot are georeferenced by differential global navigation and positioning system (DGNSS) technology. Secondly, point cloud processing and feature extraction are performed for the georeferenced TLS and ALS to form feature datasets, respectively. Thirdly, the feature-level fusion of LiDAR data from different data sources is realized through spatial join according to the tree trunk location obtained from TLS and ALS, that is, the tally can be implemented at a plot. Finally, the individual tree parameters are optimized based on the tally results and fed into the binary volume model to estimate the total volume (TVS) in a large area (whole study area). The results show that the georeferenced ALS and TLS point cloud data using DGNSS RTK/PPK technology can achieve coarse registration (mean distance ≈ 40 cm), which meets the accuracy requirements for feature-level point cloud fusion. By feature-level fusion of the two point cloud data, the tally can be achieved quickly and accurately in the plot. The proposed FFATTe method achieves high accuracy (with error of 3.09%) due to its advantages of combining different LiDAR data from different sources in a simple way, and it has strong operability when acquiring TVS over large areas.

**Keywords:** TLS; ALS; feature-level fusion; tally; volume

## 1. Introduction

As an important part of the terrestrial ecosystem, forests have a huge carbon sequestration function and are the most important carbon stock on Earth. Forests also play a special role in maintaining ecological security and coping with climate change. The TVS (Tree Volume of a Stand, m$^3$) refers to the total volume of all standing trees in a certain area and is a vital indicator of productivity and carbon storage.

The traditional TVS measurement relies on perimeter rulers, altimeters and other tools for standard plots (generally 30 m × 30 m or 50 m × 50 m) [1–4] to tally, that is, measuring the height (H) and diameter at breast height (DBH) of each tree [5,6], which will then be fed into the binary volume model to calculate the TVS [7]. However, this method is time-consuming and labor-intensive, and the measurement is subjective. Furthermore, it is not suitable for large areas due to its inability to accurately geo-locate every tree. This is unacceptable for a large-scale forest survey by satellite remote sensing, which is currently an irreplaceable means for both research and application. It requires ground

investigations to provide a large number of representative samples (number and area) in order to obtain accurate results. Light detection and ranging (LiDAR) technology can overcome the limitations of the traditional tally method and has been widely used in forestry in recent years [8]. For example, airborne laser scanners (ALS) and terrestrial laser scanners (TLS), with their high efficiency and precision, can directly measure individual tree parameters in a large area, thereby improving the efficiency of the tally [9–11].

TLS, as well as backpack and hand-held LiDAR technologies developed in recent years, can be used for ground surveys. TLS was originally mainly used for surveying and mapping with the highest accuracy and can accurately obtain individual tree parameters [12–14], such as DBH, H, etc. Henning et al. proved that TLS offers DBH measurements with an error not exceeding 1 cm and a height accuracy of <2 cm for heights of trees up to 13 m [15]. Liu et al. explored the method of using TLS to estimate the H and DBH of individual trees and found a Root Mean Square Error (RMSE) for a DBH of 1.28 cm and an H of 0.95 m [16]. Panagiotidis et al. argued that TLS significantly outperformed in DBH estimation compared with the census data, with a percent RMSE (RMSE%) of 1.9 and accuracy of 98.6%; H was slightly underestimated, with a percent bias (bias%) of $-1.9$ and an RMSE% of 5.3 [8]. It can be seen that TLS seems to have some uncertainty in retrieving H, but it can more accurately obtain the DBH in the forest.

Another rapidly emerging technology for measuring the biophysical structure of forests is ALS. ALS uses differential global navigation and positioning system (DGNSS) and inertial measurement unit (IMU) for accurate spatial positioning and obtains the vertical structure of the forest through LiDAR multiple echo technology. For instance, Andersen et al. have indicated that a tree height accuracy of $0.02 \pm 0.73$ m can be achieved from ALS data, and Kronseder et al. have also confirmed that ALS technology can assess tree height more accurately and efficiently than the field methods [17]. However, since the ALS emits laser light from top to bottom, it is difficult to penetrate the forest canopy, and it is difficult to obtain accurate DBH even in sparsely forested areas.

It can be seen from the above that TLS and ALS have their own advantages, that is, the DBH by the former is more accurate, as is the H by the latter. Once the advantages of these two types of point clouds are combined, it will definitely be more beneficial to tally. Terryn et al. proposed a method to fuse TLS and multi-echo Unoccupied Aerial Vehicle Laser Scanning (UAV-LS) to access more accurate DBH and H [18]. Panagiotidis et al. fused ALS and TLS point clouds to assess temperate managed forest structures [8]. It can be found from previous studies that point cloud registration methods are difficult to apply, especially for forests, which have complex, time-consuming procedures, high technical requirements, and low accuracy. Meanwhile, the fusion of TLS and UAV-LS is not necessary for these purposes, such as the assessment of DBH and H, etc.

In short, the accurate measurement of the forest parameters of all trees was a challenging issue due to occlusion caused by interlocking branches. There is currently a lack of methods to quickly and accurately tally plots and estimate large-area TVS, which cannot meet the needs of large-scale research and applications such as satellite remote sensing. The objective of this study is to achieve plot tally by fusing features extracted from ALS and TLS point clouds and then complete TVS estimation over a large area. Specifically, the main problems addressed in this paper include the following: (1) registering TLS and ALS point clouds in the plot; (2) tallying by fusing features extracted from different point clouds, i.e., DBH from TLS and H from ALS; and (3) estimating the TVS of large areas by optimizing the parameters of the binary volume model according to the fused features.

## 2. Materials

### 2.1. Study Area

The study area is located in the Saihanba Mechanical Forestry Field, Chengde City, Hebei province, China (42°02′~42°36′N, 116°51′~117°39′E) (Figure 1) with an elevation ranging from 1010 to 1940 m. This area is characterized by a temperate continental monsoon climate, with an annual average temperature of $-1.4$ °C and an average annual snow cover

of 7 months. The annual average precipitation is about 450 mm, with annual sunshine hours of 2368 h. The Saihanba forest farm was established in 1962, and it is currently the largest plantation in China, with a reforested area of 76,600 hm$^2$. The forest cover is 82%, of which 72.6% of the total forest area is planted.

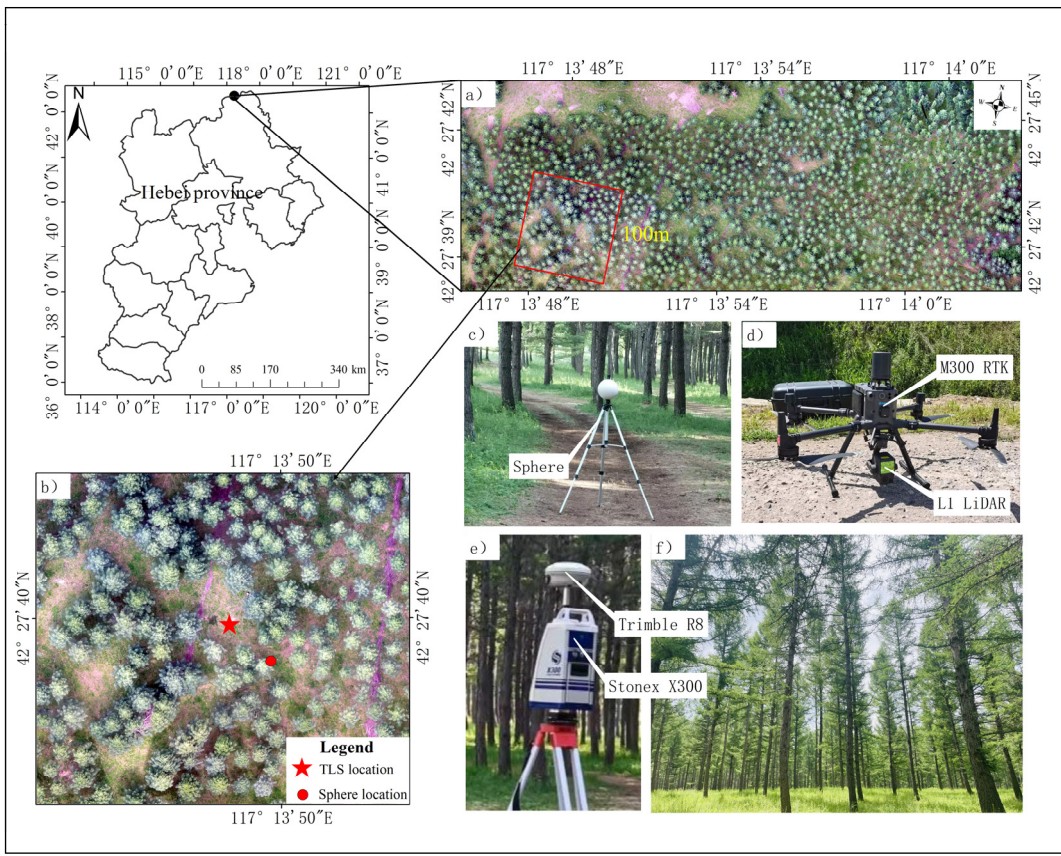

**Figure 1.** Study area. (**a**) The geographic location of the study area; (**b**) The geographic location of a plot (**c**) sphere (**d**) DJI M300 RTK multi-rotor UAV and L1 sensor (**e**) Trimble R8 and Stonex X300 (**f**) *Larix principis-rupprechtii*.

In the study area (Figure 1a), the main tree species is *Larix principis-rupprechtii*, and the understory vegetation is sparse (Figure 1f). A plot of size 100 m × 100 m (Figure 1b) in the southwest corner in the study area was chosen for tally with TLS and ALS. The TLS base station was placed in the center of the plot (★ in Figure 1b,e), and a sphere (● in Figure 1b,c) was placed at a distance of about 13 m for orienting the base station. Both the base station and the target sphere were located in the open sky area to ensure good DGNSS signals and good intervisibility between them. In this study, the area within 50 m around the TLS base station was chosen as the plot (Figure 1a) because the farther away from the TLS base station, the lower the density and accuracy of the obtained point cloud data.

### 2.2. TLS Data Acquisition and Processing

Data acquisition was conducted on 27 July 2022, when it was cloudy and breezy. TLS data were acquired using a Stonex X300 laser scanner (Figure 1e) [19–21]. Stonex X300 is a pulsed 3D laser scanner that is used for precision measurement and rapid acquisition of massive 3D point cloud data in complex environments. Its accuracy is ±4 mm@50 m, and the measurement distance is up to 300 m. Single echo and fine mode, with 360° horizontal and 180° vertical view, were employed to obtain TLS point clouds. The specific parameters of Stone X300 are shown in Table 1. The point cloud density was 5339 pts/m$^2$.

**Table 1.** Specifications of laser scanning systems.

| Technical Specifications | TLS | ALS |
|---|---|---|
| Maximum Distance Range | 300 m | 450 m |
| Range Systematic Error | 4 mm@50 m | 3 cm@100 m |
| DGNSS Precision | H: 8 mm + 0.5 ppm<br>V: 15 mm + 0.5 ppm | H: 10 mm + 1 ppm<br>V: 15 mm + 1 ppm |
| Laser Wavelength | 905 nm | 905 nm |
| Scanning Field of View | 360° × 180° | 320° |
| Scanning Speed | 40,000 pts/s | 480,000 pts/s |
| Angular Accuracy | 0.37 mrad | 0.08° |

The geographic coordinates of the TLS base station and sphere were measured using Trimble R8 (Figure 1e), by which the TLS data was georeferenced (described in Section 3.1.1). Then, the georeferenced TLS data, which mainly included ground point detection, point cloud normalization, high vegetation detection, individual tree trunk segmentation, and feature parameter extraction, were processed [22,23]. Firstly, the segmentation algorithm based on smooth surface growth was used to separate the points into clusters. Then, the irregular triangular network progressive densification filtering method was employed to filter the point cloud and perform ground point identification, followed by the generation of normalized digital surface model (CHM). Based on the CHM, the high vegetation was detected by the normalized height to the ground. Finally, a point cloud segmentation method for line entity extraction was used to extract individual tree parameters from the high vegetation points. The parameters of the mentioned above methods are shown in Table 2.

**Table 2.** Parameter settings in TLS and ALS data processing.

| Processing | Parameters | ALS | TLS |
|---|---|---|---|
| Ground point detection | Terrain inclination | 60° | 60° |
| | Iterative angle | 6° | 6° |
| | Iterative distance | 0.6 m | 0.2 m |
| Generation of CHM and high vegetation detection | Lower height value | 2 m | 0.2 m |
| | Higher height value | 80 m | 50 m |
| Individual tree trunk segmentation and feature parameter extraction | Average step length of trunk/canopy | 2 m | 0.15 m |
| | Growth step | 1 m | 0.5 m |
| | Minimum number of points contained in a single object | 20 | 40 |

Individual tree height ($H_{TLS}$) was calculated using the difference between the highest Z values recorded from the point cloud data and the Digital Elevation Model (DEM) height. Crown diameter ($CD_{TLS}$) was extracted using region growing algorithm (Solberg et al., 2006; Hyyppa et al., 2001). $DBH_{TLS}$ was measured on the stem at 1.3 m height from the ground using distance measurement function algorithm [24–26]. Finally, the extracted positions of each trunk ($Loc_{TLS}$) and $DBH_{TLS}$ were together formed into a TLS features dataset, which contains 151 trees in total. The above processes were implemented in Point Cloud Automata (PCA) v3.7.

*2.3. ALS Data Acquisition and Processing*

ALS data were collected by a DJI M300 RTK multi-rotor UAV with L1 sensor after TLS data acquisition. This ALS system has a built-in RTK module, which has a nominal positioning accuracy of 10 mm + 1 ppm horizontally and 15 mm + 1 ppm vertically. The L1 ranging accuracy is up to 3 cm@100 m, allowing for high accuracy positioning. The UAV flight height is 80 m with flight speed of 3.5 m/s and side overlap rate of 80%. Three echoes, with 320° view, were chosen to obtain high density ALS point clouds (2031 pts/m$^2$).

The same processing procedure as TLS but with different parameters was performed for ALS; the parameters are listed in Table 2. Finally, the extracted locations of each tree

trunk (Loc$_{ALS}$) were combined with H$_{ALS}$, DBH$_{ALS}$, and CD$_{ALS}$ to form the ALS features dataset. The dataset contains 166 trees in total. It should be noted that the location (x, y) of the trunk is determined according to the position of the highest point of the crown because it is not possible to obtain sufficient trunk point clouds from ALS [27]. All the above processes were implemented in PCA v3.7.

## 3. Methods

### 3.1. Feature-Level Fusion of ALS and TLS for TVS Estimation (FFATTe)

The FFATTe method mainly includes four steps, as shown in Figure 2. The first step is to obtain georeferenced TLS and ALS by high-precision positioning of both TLS and ALS through DGNSS technology; thus, the registration is realized through geographical coordinates. The second step is to perform point cloud processing and feature extraction on the georeferenced TLS and ALS to form feature datasets, respectively (referred to in Sections 2.2 and 2.3). In the third step, the spatial join is employed to perform feature level fusion according to the trunk locations in the two feature datasets to achieve tally of the plot. The fourth is to optimize the individual tree parameters depending on the tally results and feed them into the binary volume model to estimate the TVS in a large area (the whole study area). The following mainly introduces Steps 1, 3, and 4.

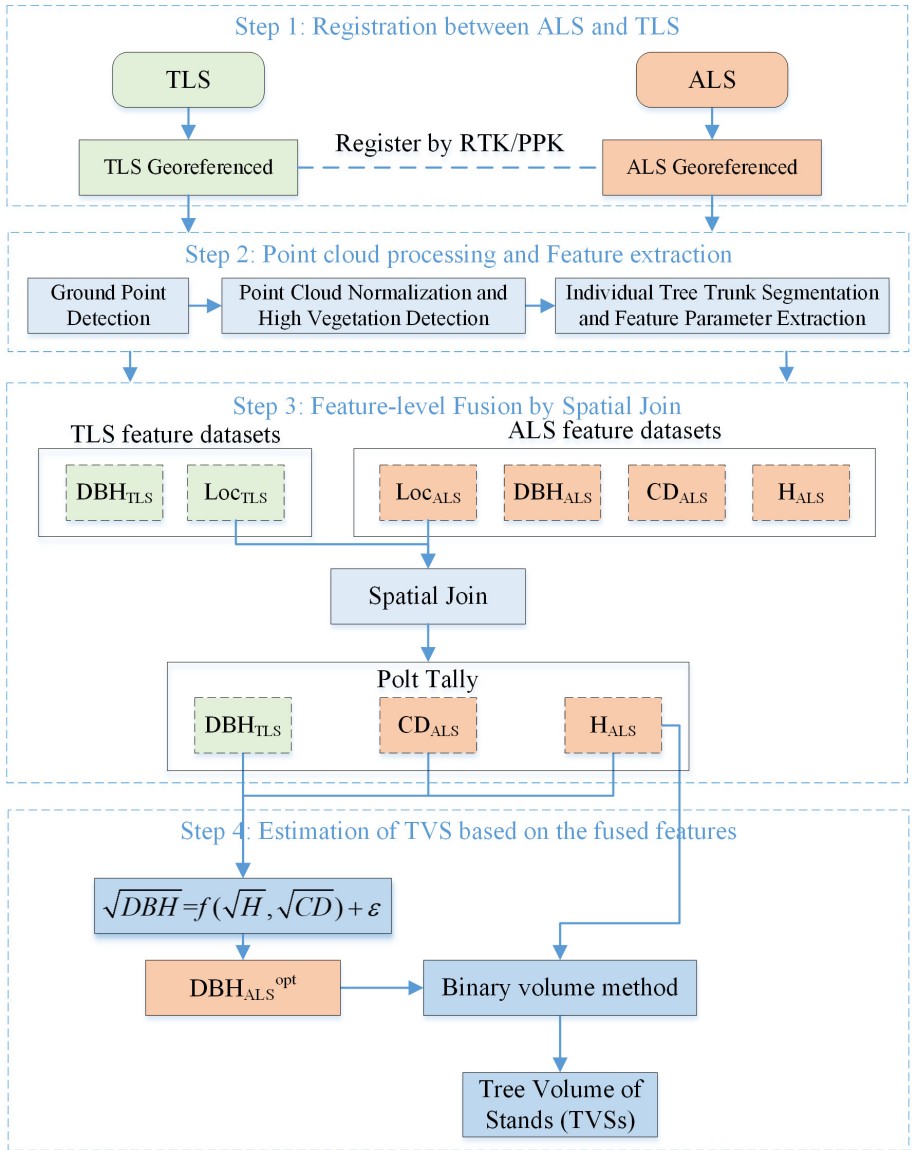

**Figure 2.** Flowchart for TVS estimation using FFATTe.

### 3.1.1. Registration between ALS and TLS

Registering ALS and TLS point cloud is the first step of the proposed FFATTe method, which is accomplished by using DGNSS technology in the data acquisition stage. Specifically, high-precision DGNSS RTK/PPK measurements are used to assign precise geographic coordinates to ALS and TLS data, respectively. Thus, the two point clouds data are registered based on the same geographic reference. In this study, we used the geographic coordinates provided by UAV IMU and DGNSS system in ALS data acquisition. The coordinate system was China Geodetic Coordinate System 2000 (CGCS2000), and the projection method was Gauss–Kruger three-degree zone projection with the central meridian of 117°E. The precision of the processed ALS data was 10 mm of horizontal and 15 mm of vertical. For TLS data georeferencing, the method of "Register by Station Location and One Point" was adopted, which means the positions of the base station and the sphere were measured, respectively, and the point cloud orientation parameters were solved through three-dimensional coordinate transformation and least square adjustment [28]. The projection and coordinate system were the same as the ALS. The precision of the processed TLS was 8 mm of horizontal and 15 mm of vertical. The data processing of ALS and TLS was implemented in DJI Terra v3.5 and StonexSiScan v3.0, respectively.

### 3.1.2. Feature-Level Fusion through Spatial Join

In this method, the feature-level fusion between the TLS and ALS feature datasets is achieved by spatial join. Specifically, according to the fact that the tree trunks in the two point clouds are very close to each other, based on $Loc_{ALS}$ and $Loc_{TLS}$ features, all other features in these two point clouds can be related by spatial join method, i.e., tallying the plot. As a result, we obtained DBH, H, and CD data of individual trees, where DBH is from TLS, and H and CD are from ALS.

The fusion process can be expressed by Formula (1):

$$C = \left\{ (a_i, b_j) \middle| a_i \in A \text{ and } b_j \in B, \ L(a_i, b_j) \leq r \right\}, \ i = 1, 2, \cdots, m; \ j = 1, 2, \cdots n \quad (1)$$

*A* and *B* represent the feature datasets of TLS and ALS, respectively, and *C* is the fused feature dataset. $a_i$ is the feature(s) of the i-th tree in *A*, *m* is the number of trees in A, and $b_j$ is the feature(s) of the *j*-th tree in *B*. *n* is the number of trees in *B*, and $j = 1, 2, \cdots n$. $L(a_i, b_j)$ refers to the distance of trunk locations extracted from TLS and ALS. *r* is search radius. The features of *A* and *B* returns the feature pairs $(a_i, b_j)$, where $a_i \in A$, $b_j \in B$, and the $L(a_i, b_j)$ are within search radius *r*.

The determination of *r* is the key to ensuring the fusion effect in this step; it was determined to be 2 m according to the distribution of tree trunk locations of the two datasets. *r* is not an adaptive value, which can match $Loc_{ALS}$ and $Loc_{TLS}$ of the individual trees to the furthest of its ability without errors if the *r* value set is over 2 m, which may match the positions of the trunk of different trees in our study area. As a result, 113 groups of $H_{ALS}$ (tree height of ALS), $CD_{ALS}$ (crown diameter of ALS), and $DBH_{TLS}$ (DBH of TLS) were obtained, i.e., 113 trees in the plot were tallied.

### 3.1.3. TVS Estimation

In this step, the binary volume model was employed to estimate the TVS(s) of a large area. Since not all DBH in the study area can be obtained by TLS and/or ALS, the DBH ($DBH_{ALS}{}^{opt}$) of trees outside of the plot can be estimated by $H_{ALS}$ and $CD_{ALS}$ using DBH optimization model. Then, $DBH_{ALS}{}^{opt}$ and $H_{ALS}$ were input into binary volume model to calculate the TVS of the study area. This section mainly includes DBH optimization model and TVS estimation for large stands.

#### DBH Optimization Model

The 113 groups of fusion results obtained in Section 3.1.2 were subjected to regression analysis to optimize DBH prediction model for outside the plot. In this way, the optimized DBH ($DBH_{ALS}{}^{opt}$) of individual trees in the whole area can be calculated only by using the

H$_{ALS}$ and CD$_{ALS}$, which means that ALS alone can be used to obtain DBH of a large stand without TLS. The Equation (2) [29] is employed to implement the above process:

$$\sqrt{DBH_{TLS}} = f\left(a\sqrt{H_{ALS}}, b\sqrt{CD_{ALS}}\right) + c \tag{2}$$

where $f$ represents iterative generalized least squares [29]. The coefficients solved in this according to Equation (2) are: $a = 0.174$, $b = 0.455$, $c = 8.871$, which are used to estimate DBH$_{ALS}^{opt}$.

TVS Estimation for Large Stands

DBH$_{ALS}^{opt}$ and H$_{ALS}$ were fed into the binary volume method (Equation (3)) [30] to accurately estimate the TVS of large stands (the whole study area, which is much larger than a plot).

$$V = \alpha * DBH^{\beta} * H^{\gamma} \tag{3}$$

where $V$ represents the volume of individual trees in the study area, and $\alpha = 0.0000942941$, $\beta = 1.832223553$, and $\gamma = 0.8197255549$ [31].

*3.2. Validation*

In order to evaluate the performance of the proposed FFATTe method, the accuracy is evaluated from three aspects: individual tree parameters, feature-level fusion effect, and TVS estimation accuracy.

For individual tree parameters, the relative differences between individual tree parameters (H, DBH, and CD) extracted from TLS and ALS are compared, respectively. For the DBH comparison in the plot, due to the TLS's high accuracy, DBH$_{TLS}$ is taken as a reference to calculate the difference between DBH$_{ALS}$ and DBH$_{ALS}^{opt}$, respectively. In addition, some statistical indicators, such as mean, maximum, minimum, standard deviation (std), and mean absolute error (ME) are calculated to evaluate the accuracy of DBH, std, and ME, as shown in Equations (4) and (5) [32,33].

$$\text{std} = \sqrt{\frac{\sum(x_i - \bar{x})^2}{n-1}} \tag{4}$$

$$\text{ME} = \frac{1}{n}\sum_{i=1}^{n} |x_i - \bar{x}| \tag{5}$$

where in Equations (4) and (5), $x_i$ is the feature value of the $i$-th tree, $\bar{x}$ is the average of all feature values, and $n$ refers to number of observations.

For feature fusion effect, both visual evaluation and quantitative analysis are used. The georeferenced ALS and TLS point clouds in the plot are superimposed and displayed in different colors, and the tree trunk locations from these two point clouds are overlapped to inspect the registration from different perspectives. At the same time, the mean, maximum, and minimum distance between the locations of TLS trunk and ALS trunk are calculated to quantitatively analyze the difference between them.

To evaluate TVS estimation accuracy, the difference between the reference, i.e., volume by DBH$_{TLS}$ and H$_{ALS}$ (V$_{ref}$), and three different volume estimation methods, namely by TLS (V$_{TLS}$), by ALS with non-optimal DBH (V$_{ALS}$), and by the proposed FFATTe (V$_{FFATTe}$), are calculated, respectively. The difference diagram of the three methods and statistical indicators (mean, maximum, minimum, and the TVS of the plot) are shown. Finally, the TVS results of the study area are mapped by FFATTe method.

## 4. Results

*4.1. Comparison of Individual Tree Parameters between ALS and TLS*

This section compares the parameters of 113 trees within the sample plot extracted from the TLS and ALS point clouds, mainly including comparisons between H, DBH, and the tree crown.

### 4.1.1. H Accuracy

Figure 3 shows the tree height of the 113 trees in the plot, with $H_{ALS}$ in green and $H_{TLS}$ in orange. The *x*-axis is the tree ID, and the *y*-axis is the height of all trees. $H_{ALS}$ is used as a reference due to the high accuracy of ALS measurements; therefore, IDs are assigned to individual trees in sequence according to $H_{ALS}$.

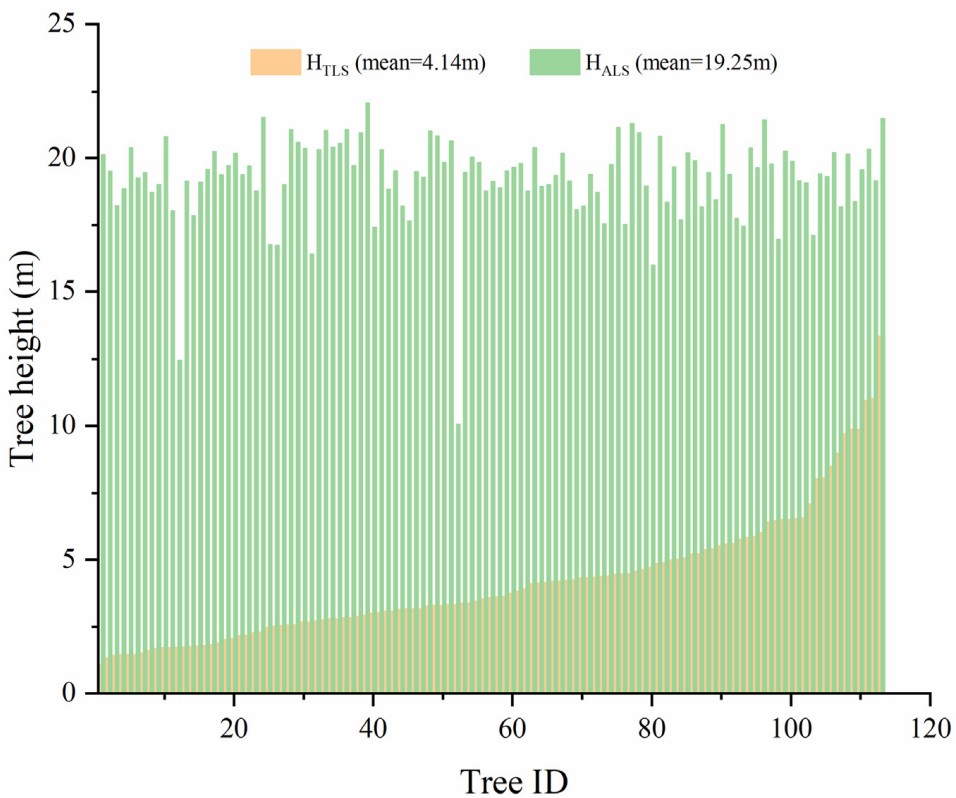

**Figure 3.** Height of 113 trees extracted by TLS and ALS.

It can be seen that the tree heights in the plot extracted by ALS are relatively close (mean = 19.25 m), which is more consistent with the field conditions. As for TLS, the extracted tree heights are significantly different from ALS, that is, the heights are lower (mean = 4.14 m), and the values vary significantly from low to high. Most of the trees are between 1 and 4 m, with only three trees taller than 10 m, which is obviously inconsistent with the field conditions.

### 4.1.2. DBH Accuracy

Three DBHs obtained in different ways are shown in Figure 4. The *x*-axis stands for the tree ID, and the *y*-axis is the DBH value. $DBH_{TLS}$ (black) is used as reference due to the high accuracy of TLS measurements; therefore, IDs are assigned to individual trees in sequence according to $DBH_{TLS}$. $DBH_{ALS}$ (red) is estimated by using the non-optimized DBH [31], while $DBH_{ALS}^{opt}$ (blue) is the optimized result obtained by using the fused features (plot tally) (Section DBH Optimization Model). It should be noted that both $DBH_{ALS}$ and $DBH_{ALS}^{opt}$ are not directly extracted from the ALS point cloud but estimated by using the tree crown diameters and tree heights.

It can be seen from the reference ($DBH_{TLS}$) that the DBH of the trees in the plot varies widely, from the minimum 12.90 cm to the maximum 34.23 cm. The two DBH estimation methods using ALS have certain deviations, and the deviations have certain rules. According to the $DBH_{TLS}$ partition, when $DBH_{TLS}$ is between 12.90 cm and 22.06 cm, both $DBH_{ALS}$ (ME = 5.31 cm) and $DBH_{ALS}^{opt}$ (ME = 3.67 cm) are overestimated, and the over-estimation of the former is more serious. When $DBH_{TLS}$ varies between 22.06 cm and 27.11 cm, $DBH_{ALS}$ has almost always been overestimated, while $DBH_{ALS}^{opt}$ is overesti-

mated in low-value areas and underestimated in high-value ones. The overall deviation of $DBH_{ALS}^{opt}$ (ME = $-0.67$ cm) is much smaller than $DBH_{ALS}$ (ME = 1.28 cm). When $DBH_{TLS}$ is in the range of 27.11 cm and 34.23 cm, compared with $DBH_{ALS}$ (ME = $-2.86$ cm), the underestimation of $DBH_{ALS}^{opt}$ (ME = $-5.39$ cm) is more obvious. Actually, the distribution of DBH of trees conforms to the normal distribution, and the median value accounts for a large proportion. Generally, the DBH estimation model fitted by CD and H has higher accuracy in the median, which may cause $DBH_{ALS}^{opt}$ and $DBH_{ALS}$ to be overestimated when $DBH_{TLS}$ is smaller but underestimated when $DBH_{TLS}$ is larger. Overall, the percentage errors of $DBH_{ALS}^{opt}$ and $DBH_{ALS}$ are 0.30% (ME = $-0.07$ cm) and 8.00% (ME = 1.89 cm), respectively. $DBH_{ALS}^{opt}$ has a better estimation effect on DBH than $DBH_{ALS}$, with a 7.70% reduction in error.

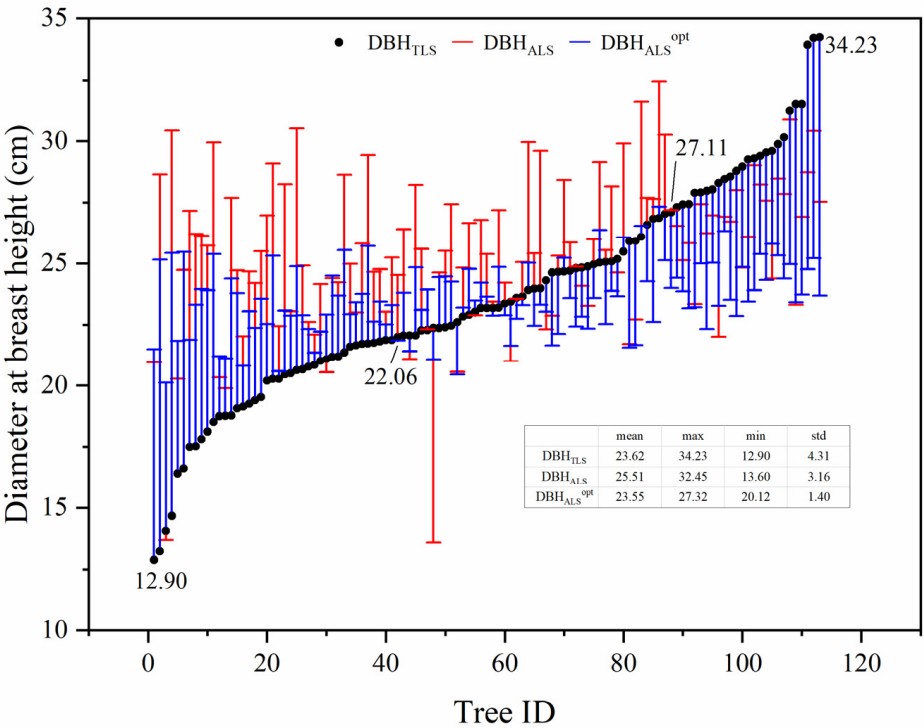

**Figure 4.** DBH of 113 trees extracted by TLS and ALS.

From the perspective of standard deviation, the reference is the largest (4.31), while $DBH_{ALS}$ and $DBH_{ALS}^{opt}$ are lower, with 3.16 and 1.40, respectively. Although the standard deviation of $DBH_{ALS}$ is closer to the reference, it can be seen from Figure 4 that for each individual tree, compared with the optimized DBH ($DBH_{ALS}^{opt}$), the unoptimized DBH ($DBH_{ALS}$) is more different from the reference. The unoptimized method severely affects the accuracy of tally. Therefore, it is necessary to optimize the DBH estimation model using $DBH_{TLS}$ to improve the DBH estimation accuracy.

### 4.1.3. Crown Accuracy

The tree crown diameters extracted by TLS ($CD_{TLS}$, orange) and ALS ($CD_{ALS}$, green), respectively, are shown in Figure 5. The x-axis is the tree ID, and the y-axis is the crown diameter of all trees. IDs are assigned to individual trees in sequence according to $CD_{ALS}$.

It can be seen that the range of $CD_{TLS}$ is 0.54 m to 4.85 m, and its maximum value is only slightly higher than the minimum value of $CD_{ALS}$ (4.44 m). Furthermore, the $CD_{TLS}$ (mean = 2.63 m) is significantly smaller than $CD_{ALS}$ (mean = 8.08 m) due to the inability to obtain a complete canopy point cloud with single-station TLS mode (Figure 6b). According to Figure 5, the $CD_{ALS}$ is more accurate, while $CD_{TLS}$ is obviously inconsistent with the ground truth (actual tree crown diameters), which can be seen from the crown image in Figure 6a.

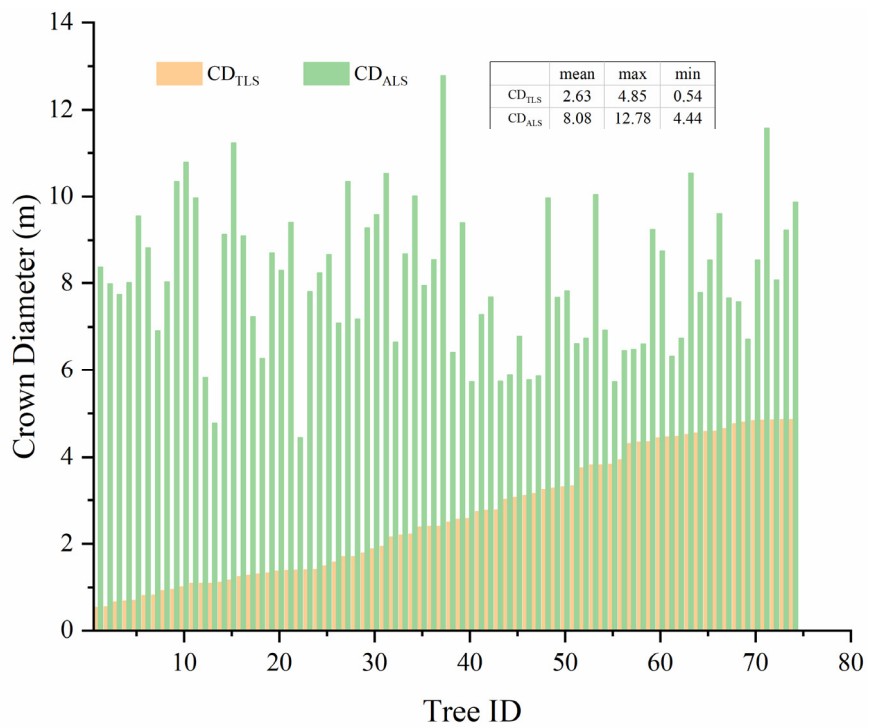

**Figure 5.** Crown diameter of 113 trees extracted by TLS and ALS.

## 4.2. Individual Tree Trunk Location

Figure 6 shows the difference in trunk location extracted by TLS and ALS, as well as the point cloud details of several individual trees.

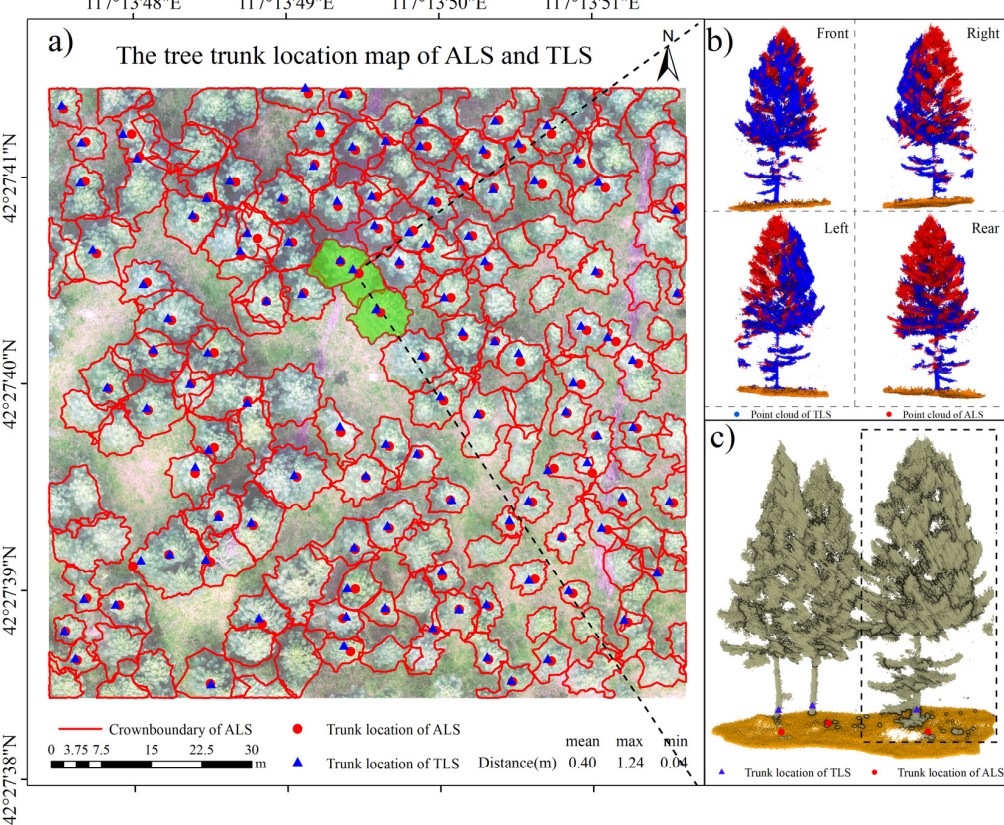

**Figure 6.** (**a**) Position differences of tree trunks extracted by TLS and ALS; (**b**) the point cloud of ALS in red and TLS in blue; (**c**) the trunk location of three trees.

In Figure 6, ● and ▲ stand for the trunk locations by ALS and TLS, respectively. In Figure 6a, the red polygon is the crown boundary extracted by ALS. Figure 6b shows the registration details of the point clouds from different perspectives of a tree from Figure 6c. Figure 6c shows the trunk locations of three trees from the study area obtained by ALS and TLS.

It can be seen that the statistical results are basically consistent with Figure 6a. The mean distance of trunk positions extracted from the two point clouds is 0.40 m, which accounts for 4.96% of the mean crown diameter extracted by ALS. Figure 6 also shows that the trunks of the TLS and ALS point cloud data almost coincide. This indicates that the overall difference in trunk position deviation between the two point clouds is not very large, which meets the requirement of feature-level registration.

As can be seen from Figure 6c, the trunk positions extracted by TLS are almost exactly the same as the actual positions obtained by ALS scanning, while the position predicted by ALS is quite different from it. It can be seen that the results obtained by using crown vertices to determine the trunk position are not accurate. It is necessary and feasible to use TLS trunk information to supplement and optimize the trunk of ALS.

### 4.3. Estimation of TVS

The volume ($V_{ref}$) calculated by $DBH_{TLS}$ and $H_{ALS}$ as the input parameters of the binary volume model is taken as a reference to evaluate the accuracy of the other three methods. Figure 7 is the comparison chart of the volume obtained by the four methods.

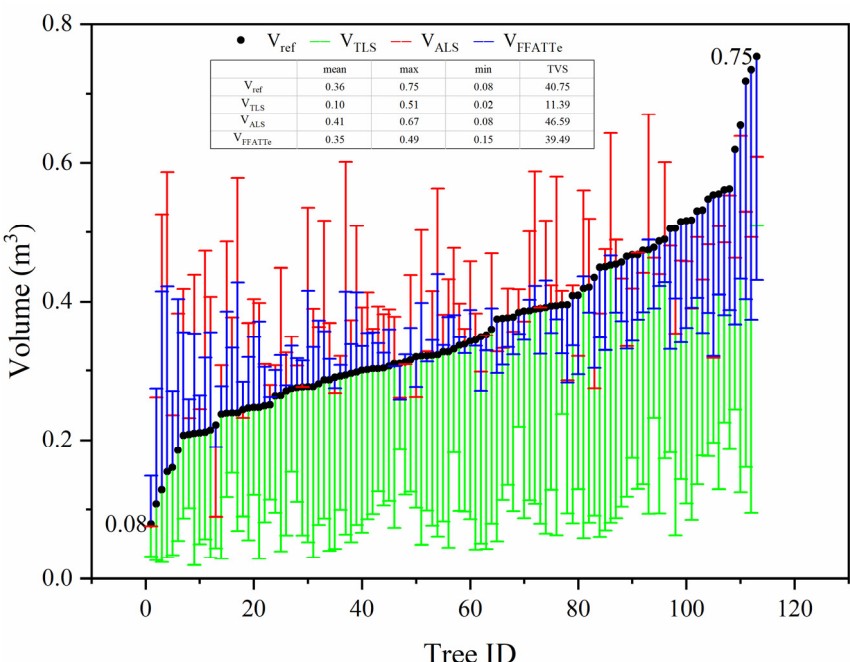

**Figure 7.** Individual tree volume results obtained by four methods ($V_{ref}$, $V_{TLS}$, $V_{ALS}$, and $V_{FFATTe}$).

As is shown in Figure 7, the *x*-axis is the tree ID, and the *y*-axis is the volume of individual tree. Green is the volume ($V_{TLS}$) calculated using TLS only. Red refers to the volume ($V_{ALS}$) calculated using unoptimized parameters. Blue is the volume ($V_{FFATTe}$) calculated with optimized parameters, i.e., the proposed FFATTe method. Black stands for the reference ($V_{ref}$).

As can be seen from the above figure, the reference volume ($V_{ref}$) varies from 0.08 m$^3$ to 0.75 m$^3$. $V_{TLS}$ is grossly underestimated on almost every individual tree. Similar to the changing trend of DBH (Section 4.1.2), $V_{ALS}$ and $V_{FFATTe}$ also move from low-value overestimation to high-value underestimation as the $V_{ref}$ increases. Additionally, $V_{ALS}$ differs more from the reference than $V_{FFATTe}$.

The statistical results are basically consistent with the performance in the figure (Figure 8). For the total volume (TVS), the reference TVS is 40.75 m$^3$, and the errors of

$TVS_{TLS}$, $TVS_{ALS}$, and $TVS_{FFATTe}$ are 29.63 m³ (72.05%), 5.84 m³ (14.33%), and 1.26 m³ (3.08%), respectively. It can be seen that TLS significantly underestimates TVS and has the worst estimation accuracy, which is mainly because the tree height calculated by TLS is inaccurate (Section 4.1.1). The FFATVe method has the highest estimation accuracy for TVS. Compared with $TVS_{ALS}$ calculated with unoptimized parameters, the percentage error of $TVS_{FFATTe}$ is reduced by 11.25%. The results show that the proposed FFATTe method can significantly improve the accuracy of TVS.

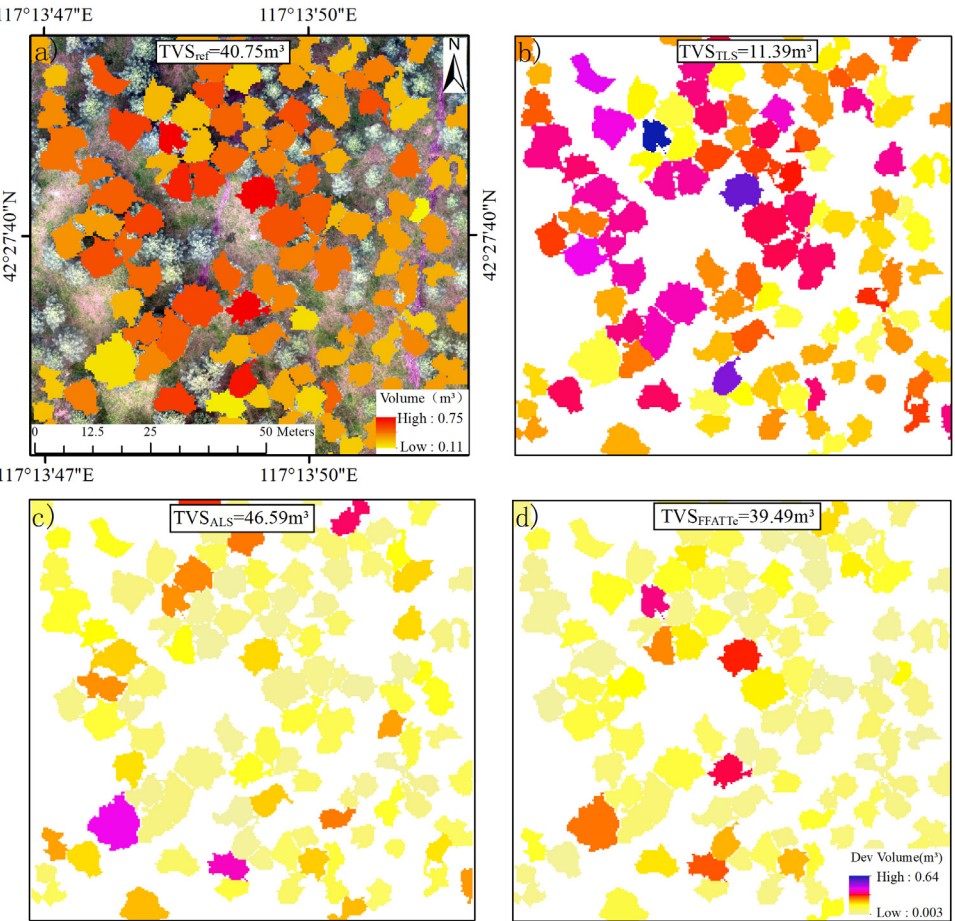

**Figure 8.** (**a**) the referenced volume extracted by TLS and ALS; (**b**–**d**) represent the difference plots between three methods and the reference, respectively.

Figure 8 shows the reference volume and the difference between the three methods and the reference in plot scale.

Figure 8a shows the reference volume calculated by TLS and ALS. Figure 8b–d represent the difference between $V_{TLS}$, $V_{ALS}$, $V_{FFATTe}$, and $V_{ref}$, respectively. The darker the color, the larger the difference between volume and reference. In Figure 8b, purple and blue are the most prominent, while in Figure 8d, yellow is the most prominent, and purple and blue are the least prominent. This shows that the difference between $V_{TLS}$ and the reference is the largest, while that of $V_{FFATVe}$ is the smallest. Therefore, the FFATTe method can greatly improve the estimation accuracy of volume, which is consistent with the previous conclusions.

Figure 9 is the TVS result of the study area calculated by the proposed FFATTe method.

The FFATTe method was applied to the study area to obtain the volume result of large area. Finally, the feature parameters and TVS data of 1197 trees were obtained. The mean values of the tree height, DBH, and crown diameter are 18.27 m, 23.53 cm, and 8.28 m, respectively. The mean volume and TVS are 0.34 m³ and 400.93 m³, separately.

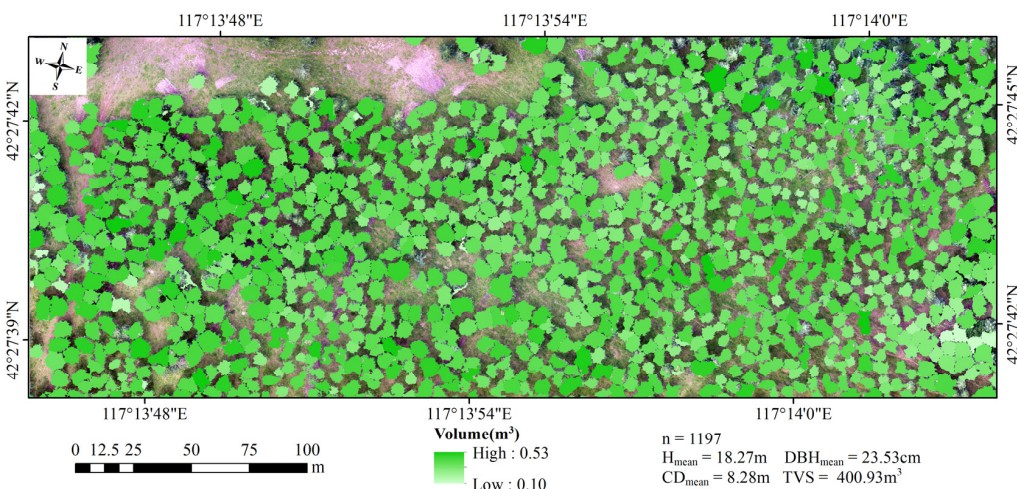

**Figure 9.** The volume result of study area.

## 5. Discussion

### 5.1. Point Cloud Registration and Features Fusion

This study has proposed a new method of fusing ALS and TLS point cloud data for forest inventory on a single tree level. In this paper, the precise measurement of each tree can be achieved without the high-precision registration of TLS and ALS point clouds. This means measurements of basic forest parameters for estimating TVS could be obtained more efficiently. Although there are many proposed registration methods so far [34–36], most of them are oriented to high-precision applications, such as mapping and monitoring a compound slow-moving rock slide [37], three-dimensional (3-D) reconstruction of buildings [38], robust assessment of tree crown structure [39], etc. These methods can achieve high accuracy of registration, but they are complicated, time-consuming, and difficult to apply. However, it is not necessary for applications that do not require high accuracy, such as for the estimation of tree volume of stands in this paper, as long as it is possible to distinguish which tree the measured data belongs to, that is, to achieve the measurement of each tree (tally).

The idea of feature-level fusion proposed in this study is to achieve a tally in the plot, that is, to fuse the H extracted from ALS and the DBH from TLS for each tree. This means that the parameters of each tree can be obtained without a lot of manpower and material resources. At present, the tally is often done by numbering trees in forest surveys, but it is difficult to correspond the survey results with remote sensing data. Many researchers have tried tallying with remote sensing or surveying technology. For example, Berra et al. (2019) measured the coordinates of each tree using GPS-RTK [40], and Terryn et al. set up a DGNSS base station outside the forest to obtain the geographic coordinates of a ground control point and then used it as a benchmark to measure the location of each tree under the forest with a total station [18]. However, in the dense forest, the disadvantages of these methods are obvious. First, the DGNSS measurement accuracy is low or even impossible due to weak signals, and second, it is labor-intensive, especially for large plots.

In this study, we used GNSS-RTK to georeference the TLS and ALS point clouds, respectively, and the registration between them was accomplished according to the geographic coordinates. This method is especially applicable for forests, as it can stay away from the registration challenges caused by the inconsistency of perspectives and densities between these two point clouds. In addition, to verify the validity of accurate registration for the feature-level point cloud fusion method. We precisely registered these two point clouds with ICP [41], and the distance between them was reduced from 40 cm to around 20 cm. Although a 50% improvement looks great, it does not substantially improve the subsequent feature-level fusion but increases processing steps and uncertainty. Therefore, the accuracy of this registration method is sufficient for feature-level fusion, and it can achieve tallying at a low cost.

To achieve feature-level fusion, we spatially joined the extracted features by trunk positions from TLS and ALS, respectively. The former can be directly extracted by TLS, but the latter can only be estimated due to the low number of ALS echoes and insufficient penetration. In this paper, the position (x, y) of the highest point of a tree is taken as its trunk location [42–45]. They both fall under the crown of the same tree (Figure 6a). The distance between them is very small, which only accounts for 4.96% of the mean tree crown diameter and has little effect on feature-level fusion. Moreover, previous fusion mainly referred to the fusion of the same parameter, while in our study, the fusion process is more about parameter selection. Therefore, the tally can be realized by employing spatial join, which can accurately combine the DBH of TLS and the corresponding H of ALS. With the development of LiDAR technology, it will be possible to obtain the trunk positions directly from ALS using more powerful multi-echo LiDAR (e.g., 7 or 14 echoes, etc.) [46,47]. By then, many of the technical difficulties in current forest investigation will be resolved, and forest volume estimation will be more efficient.

The limitation of the proposed FFATTe is that it must georeference the point cloud data using DGNSS, which is usually difficult for TLS data acquisition. As we all know, it is difficult to find open sky in the dense forest, which is necessary for DGNSS measurement. One way to solve this problem is to employ LiDAR with a long range. For example, the range of TLS in this paper is up to 300 m, while the general plots are set to 30 m or 50 m. Therefore, the station can be set at the edge of the forest, and small plots deep in the forest and far from the edge can be taken. Although the density and accuracy of point clouds at further distances are relatively reduced, the accuracy is sufficient for high-precision LiDAR. The other way is to use handheld and/or backpack LiDAR, which can accomplish georeferencing by calculating the distance and attitude change of its relative motion. Pierzchała et al. (2018) mapped forests using an unmanned ground vehicle with 3D LiDAR and graph simultaneous localization and mapping (SLAM) and found that robust SLAM algorithms can support the development of forestry by providing cost-effective and acceptable quality methods for forest mapping [48]. However, these two technologies have difficulty obtaining high-precision positioning [49], and the extracted DBH accuracy is also relatively low at present [50]. Moreover, the measurement path needs to be planned in advance, and the implementation process is complicated [51–54]. With the development of positioning technology in the future, the FFATTe method will become more flexible and convenient. For example, if indoor positioning technology can be applied to forest surveys, the location of base station will not have to be considered.

*5.2. Individual Tree Parameters and TVS*

Although there are many excellent and accurate forest volume estimation methods, for example, unary volume method, binary volume method, ternary volume method, deep learning method, LiDAR and Hyperspectral combined method, etc. [55–57]. However, some of them require a large number of input parameters, resulting in poor practicability; although some of them are simple, they cannot effectively utilize various sampling data.

In this paper, the binary volume model is used, which needs input H and DBH to estimate TVS. As far as the current remote sensing means are concerned, the H of ALS and the DBH of TLS are the most accurate and easy to obtain. For H, Wang et al. (2019) found that ALS is the most accurate measurement method [58], while for DBH, since ALS cannot directly obtain DBH, TLS has become the only option at this stage. Although DBH can be estimated from other parameters obtained from ALS, such as H and/or crown diameter [29], the accuracy is not satisfactory. Jucker et al. estimated DBH with H and crown diameter, respectively and jointly, and found that the RMSEs were 13.7 cm, 16.6 cm, and 9.7 cm [59]. Combining crown diameter and H to estimate DBH greatly improves the estimation accuracy of DBH. However, such methods will still reduce the accuracy of the binary volume model. Therefore, we optimized the DBH estimation parameters based on the fused accurate individual tree data, resulting in a 7.70% increase in the percentage error of DBH and a 11.25% of volume accuracy. Moreover, for the DBH estimation of trees

outside the plot, the tally data obtained from the plot must be used for optimization, by which the accurate tree volume could be achieved for other stands in a large area. In short, the binary volume model we adopted in FFATTe is the simplest and most effective way to estimate TVS at this stage, which can make full use of the highest precision data from different sources, i.e., the DBH of TLS and H of ALS.

The result of the crown in this study is also worth mentioning. Panagiotidis et al. has shown that the crown diameter extracted from TLS is accurate and the RMSE is only 0.5 m [8]. However, this is not consistent with our results, which show that TLS has a poor ability to extract the crown, with an error of 67.45% in diameter. Novotny et al. also obtained the same result as ours [60]. It can also be seen from Figure 7 that the estimated volume accuracy by using only TLS data is the worst. There are two possible reasons for this. First, those studies used multi-station scanning stitching, so complete tree information could be obtained from different perspectives. Second, the trees around the TLS base station are too high, so the high tree canopy cannot be reached while the distant trees are blocked by the others, which leads to their canopy not being able to be fully scanned. Therefore, it is impractical to tally only by TLS in a plot. It was obvious that, from the visual effect (Figure 6a), the crown extracted from ALS is in good agreement with the actual tree, which was consistent with the results of Liu et al. (2022) [60]. In addition, the tree canopy may be more valuable for large-scale forest volume estimation using satellite remote sensing, with which it is difficult to obtain DBH and H but easier to extract the tree canopy [61,62]. Our study found that there is a relative strong correlation between crown diameter ($CD_{ALS}$) and individual tree volume ($R^2 = 0.45$), which may provide the possibility for forest volume estimation using satellite remote sensing technology in the near future [63].

## 6. Conclusions

In this paper, a FFATTe method is proposed for the tree volume of stands estimation for large areas by fusing features extracted from TLS and ALS point cloud data, respectively. This method firstly registers TLS and ALS point cloud data in a small plot by georeference, respectively. Then, the fused feature dataset, which is generated by spatial join and contains the individual tree parameters in the plot, is used to optimize ALS-based DBH estimation for outside the plot. Finally, both the optimized DBH ($DBH_{ALS}^{opt}$) and the tree height ($H_{ALS}$) extracted from ALS are sent into the binary volume model, through which the tree volume of stands in a large area can be estimated.

The main conclusions of this study are as follows: (1) the georeferenced ALS and TLS point cloud data using DGNSS RTK/PPK technology can achieve coarse registration (mean distance $\approx 40$ cm), which meets the accuracy requirements for feature-level point cloud fusion; and (2) the feature-level fusion of the two point clouds can be achieved by spatial join according to the tree trunk location extracted/estimated from TLS/ALS. In other words, LiDAR of different platforms can be integrated together to tally (measure each tree) in a plot, and the results can optimize the model for DBH estimation from ALS (with 7.70% reduction in percentage error); and (3) the proposed FFATTe method achieves high accuracy (with error of about 3.09%) due to its advantages of combining different LiDAR data in a simple way, which has strong operability and practicability for large-area TVS estimation.

This method has only been verified in a managed coniferous forest, which are all *Larix principis-rupprechtii* that are tall and straight with similar growth. However, there are no experiments on the more complex situations, e.g., broadleaved forests, mixed forests, and complex understory vegetation. For these cases, the FFATTe method itself does not need to be changed, but individual tree feature extraction will face challenges. For example, overly dense understory vegetation may affect both DBH extracted from TLS and tree height from ALS. Therefore, more attention should be paid to those methods of combining multi-source data to improve the accuracy of individual tree feature extraction.

**Author Contributions:** Conceptualization, L.G. and L.D.; methodology, Y.W. and L.G.; software, L.G. and Y.W.; validation, L.D., L.G. and Y.W.; formal analysis, L.G.; investigation, L.G.; resources, Y.W.; data curation, L.G.; writing—original draft preparation, L.G.; writing—review and editing,

Y.C.; visualization, J.Z.; supervision, P.H.; project administration, L.D.; funding acquisition, L.D. All authors have read and agreed to the published version of the manuscript.

**Funding:** This study was funded by the R&D Program of Beijing Municipal Education Commission (No. KZ202210028045) and Special Project of High-Resolution Earth Observation System (HREOS) (No. 05-Y30B01-9001-19/20).

**Data Availability Statement:** Data will be made available upon request.

**Acknowledgments:** We would like to thank Zhuo Lu and Kongbo Wang for their valuable support.

**Conflicts of Interest:** The authors declare that they have no known competing financial interest or personal relationships that could have appeared to influence the work reported in this paper.

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
