# Peer review of "A Feature-Level Point Cloud Fusion Method for Timber Volume of Forest Stands Estimation"

_remotesensing, doi:10.3390/rs15122995_

Round 1

Reviewer 1 Report

A brief summary

The article proposes a feature-level point cloud fusion method called FFATTe. This method involves georeferencing the TLS and ALS point cloud data, performing point cloud processing and feature extraction, and then fusing the data through spatial join to achieve accurate three-dimensional measurements. The individual parameters are then optimized and used to estimate the total volume. The authors claim that the FFATTe method is highly accurate and has strong operability for large-scale applications.

General concept comments

Abstract

The abstract is too long and should be shortened. It should only include relevant information about your method, avoiding unnecessary details and background information. This will help ensure that the abstract is concise and easy to read, while still providing readers with a clear understanding of your approach and its significance.

Section 5: 

Section 5 (Discussion) is inadequately written and needs to be rewritten. It is difficult to understand what the discussion is about. I strongly feel that the first two paragraphs (lines 384-395 and 396-404) should belong to Section 1, where existing approaches are presented. The third and fourth paragraphs, on the other hand, should be included in the chapter where the results are presented, as some discussion about various graphs and results has already been performed.

Specific comments 

Page 3, Line 107-108:

»The dominant tree species are Larix principis-rupprechtii, Picea asperata, and Betula platyphylla, etc.«

You should provide photographs of these tree species that your method can measure, and highlight their specific characteristics to help readers better understand them.

Page 5, Line 148:

What is the abbreviation for PCA?

Page 5, Line 162:

What is the abbreviation for PCC? This PCC has the same version (v.3.7) as PCA?

Page 7, Line 228, Equation 2:

Equation 2 is unclear because the meaning of function f is not explained. What does function f represent, and what are the coefficients a, b, and c used for?

Page 8, Line 228, Equation 3:

Equation 3 also has coefficients a, b, and c, but with different values. Is there any connection between these coefficients and the coefficients in Equation 2?

Page 10, Fig 4.

How exactly are the three sorted? Do they always have the same IDs in all figures?

Can you explain why DBH_ALS^OPT and DBH_ALS are overestimated when DBH_TLS is smaller, but underestimated when DBH_TLS is larger?

Page 11, Lines 315-316:

»Fig. 5, the CDALS is more accurate, while CDTLS is obviously inconsistent with the ground«

I can see what the ground truth is in Figure 5. Could you please clarify which ground truth you are referring to?

Page 17, Lines 513:

»This method has only been verified in a managed coniferous forest, which are all Larix principis-rupprechtii, tall and straight with similar growth. However, there is no experiments on the more complex situations,«

On page 3, lines 107-108, you mention other tree species, but in this section, you stated that your method has only been verified on one tree species. Is a tall and straight condition with similar growth ideal for your method? I am afraid that more experiments and verification should be conducted to support your claim.

Overall, the grammar and style of the article are good, but there is still room for improvement with the help of a native speaker.

Author Response

Dear reviewer,

We gratefully appreciate the editors and all reviewers for their time spend making positive and constructive comments. These comments are all valuable and helpful for revising and improving our manuscript entitled “A feature-level point cloud fusion method for timber volume of forest stands estimation (ID: remotesensing-2389391), as well as the important guiding significance to our researches.

We have studied comments carefully and have made correction which we hope meet with approval. Revised sections appear in revised manuscripts in revision mode. Point-by-point responses to the three nice reviewers are listed the Revison Report.

Reviewer 2 Report

(1)The detail of feature-level fusion is not clear.

(2)The difference of Tree Parameters by ALS and TLS is significant. Is it effective to select them as the features of fusion?

(3)As mentioned in the paper, “The determination of ‘r’ is the key to ensure fusion effect, ……”, why it is 2m, not a adaptive value? How does the error of  ‘r’ affect the results?

Author Response

(The authors gave the same response as above.)

Reviewer 3 Report

The paper seems to suit Remote Sensors scope. The paper is focused on registering laser scanning point clouds, tallying by fusing features extracted from point clouds mentioned and estimating tree volume of a stand of large area by optimizing the parameters of the binary volume model according to the fused features. The level of originality of the paper is good and the paper is interesting and quite well written. In Introduction, the authors highlighted the paper’s objective and gave the topic background with some relevant references. The methodology is described sufficiently; however, the authors should add some necessary references there. The authors presented the results and discussion in sufficient way. The conclusions are consistent with the results and the question addressed.

There are my detailed comments to article reviewed:

1.    Page 1, lines 44-45. Despite of providing full names of H and DBH in the abstract, I recommend repeating them in the main text when the acronyms appear for the first time.

2.    Page 2, lines 63, 65-66. Please provide full names of RMSE and RMSE% and describe shortly these measures as well as bias%. Very similar remark to Page 8, line 246 – Please describe shortly standard deviation and mean error (e.g., the way of computing them).

3.    Page 2, line 81. Please provide full name of UAV-LS. Please check if the full names are provided for all acronyms within the text in the whole paper.

4.    Page 3, line 107-108. I suppose that Latin names of trees should be written in Italics. The same remark concerns the rest of the manuscript.

5.    Page 6. The placement of Fig. 2 is confusing. Within the text the authors refer to it on Page 4, while the flowchart is discussed in Section 3.1 on Page 6.

6.    Pages 6-8. Please add some references to Section 3 “Methods” where it is possible.

7.    There are a lot of typos within manuscript, especially no space between number and unit, no Italics in Section 3.1.2, no superscripts in units as “hm2” or “m3”.

I recommend minor revision of the paper before the publication in Remote Sensing.

Author Response

(The authors gave the same response as above.)

Round 2

Reviewer 1 Report

I do not have further questions and comments.

I do not have further comments.